



# Method for using spectral flow data to predict vortex-induced vibration onset of static structures

Kevin R. Moore[1], Hannah K. Ross[2], Kirk L, Bonney[1], and Brent J. Summerville[2]

[1]Sandia National Laboratories [*], PO Box 5800 MS 0717 Albuquerque, NM 87185, USA
[2]National Laboratory of the Rockies, 15013 Denver West Parkway, Golden, CO 80401, USA

**Correspondence:** Kevin R. Moore (kevmoor@sandia.gov)

**Abstract.** Spectral representations of vortex shedding behavior, such as airfoil data derived from computational fluid dynamics or experiment, can support the efficient identification of potential vortex-induced vibration onset. However, use of such data is hindered by the myriad of practical considerations required to scale, filter, and rank the risk of overlap between natural frequencies and shedding frequencies. This problem spans across Reynolds number, local angle of attack, and local skew angle, in addition to frequency harmonics, interpolation, and multi-body multi-element structures. The combinatorial scale of the problem additionally necessitates efficient numerical methods. This paper presents a reproducible, open-source framework with accompanying source code and a graphical user interface. With the improvements here, these problems can be addressed to enable the straightforward use of existing spectral datasets for arbitrary beam-type structures. We describe the methods, present a simple verification case, and exercise the method on a sample structure using a spectral airfoil dataset. The framework enables designers to readily navigate the complex space to identify, avoid, and include (via one-way coupling) the onset of vortex-induced vibration in their design workflows. Code, example datasets, and reproduction assets are openly released.

## 1 Introduction

Static structures such as communication towers, cables, parked wind turbines, and marine structures can experience vortex-induced vibration. When flow separates periodically in a fluid wake, alternating vortices impose fluctuating forces. If the shedding frequency coincides with a natural structural frequency, large-amplitude oscillations can result. While vortex shedding from cylinders is well characterized (Roshko, 1961; Schewe, 1983), shedding from shapes between a cylinder and a flat plate, such as airfoils, contains more complex spectral forcing harmonics, which complicates use for design. This can be especially important for certain types of wind energy structures, including vertical-axis wind turbines (Sakib and Griffith, 2022), and large horizontal-axis wind turbines (Grinderslev et al., 2023). These structures leverage airfoils that produce bluff-body shedding in some regimes.

For airfoils, Sheldahl and Klimas (1981) compiled polar data for symmetric NACA sections up to an angle of attack (AoA) of 180°. At high AoAs, this dataset approached flat-plate behavior. Unsteady pressure measurements by Swalwell et al. (2003) on

---

[*]Sandia National Laboratories is a multimission laboratory managed and operated by National Technology and Engineering Solutions of Sandia, LLC., a wholly owned subsidiary of Honeywell International, Inc., for the U.S. Department of Energy's National Nuclear Security Administration under contract DE-NA-0003525





a NACA 0021 demonstrated multiple distinct vortex shedding peaks that decreased in frequency with AoA, whereas Strouhal numbers based on projected chord remained nearly constant and consistent with flat-plate behavior, like that characterized by Chen and Fang (1996). More recently, Benner et al. (2019) conducted an experimental campaign using the NACA 0021 in water and observed significant energy across multiple Strouhal harmonics depending on AoA, but the results were generally consistent with prior work.

Modern computational studies have extended this kind of study to other airfoils. Bidadi et al. (2024) performed three-dimensional improved delayed detached-eddy simulations of FFA-W3 airfoils over the full 0–360° AoA range. They showed Strouhal numbers collapse to nearly flat-plate behavior for high AoAs but for a continuum of airfoil thickness between a flat plate and cylinder. However, for low AoAs, or near 180°, there is greater variation. Further investigation into this dataset also shows multiple frequency peaks for many conditions and confirms complexities identified by Fernandez-Aldama et al. (2025).

There is now a significant body of information suggesting that 1) vortex shedding of airfoils generally follows flat-plate behavior, 2) Reynolds effects may be minimal for non-transition conditions at high angles, 3) there is a continuum of Strouhal numbers between a flat plate (lower) and cylinder (higher) based on thickness, and 4) there is some variance in Strouhal numbers with AoA at high AoAs, and significant variance in Strouhal numbers at low AoAs. With this knowledge, the next step is to develop a practical process, spanning the highly dimensional problem, that uses the data for design with greater precision.

## 1.1 Objectives

While this topic is relevant to a broad variety of applications, for this paper, we will focus on the example case of a small, vertical-axis distributed wind turbine. The case is chosen because this structure has a clear overlap potential between shedding frequency and structural natural frequency. For the reference turbine described in Moore et al. (2024), the first three major blade-strut system natural frequencies are approximately 1.8, 4.5, and 5.7 Hz. A cylinder's Strouhal number just below the cut-in inflow velocity at 4.5 ms$^{-1}$ is approximately 0.2 (Roshko, 1961), and for a flat plate it ranges between 0.2 and 0.8 depending on AoA (Chen and Fang, 1996). This simplified analysis puts the shedding frequency range between 1.8 Hz and 7.2 Hz, or directly on top of the blade natural frequencies for certain parked angles and low-velocity inflow conditions.

Failing to design around the potential shedding overlap or to account for the additional fatigue cycles poses a significant risk to this type of structure or other structures with similar properties. Additionally, these types of shapes do not produce only a single shedding frequency in the wake, and vibration can lock into different harmonics depending on initial conditions (Grinderslev et al., 2022). Prior work (Bidadi et al., 2024; VimalKumar et al., 2024) has effectively shown how to apply the dominant frequency to highlight one problem area, but there are other risks not covered by this initial breakthrough work. These risks include 1) the effect of the drag vector's frequency content, 2) built-up structures' range of AoAs for a given inflow condition, 3) modal harmonics, 4) non-primary shedding frequency impacts, and 5) how to reliably traverse sparse data through interpolation. These are the main contributions this paper aims to make.





## 1.2 Paper structure

The paper is organized as follows: Sect. 2 describes the methods used, including spectral data decomposition, linear algebra to resolve multi-body AoAs, interpolation in both Reynolds and Strouhal domains, and combinatorial reduction. Sect. 3 introduces the VorLap code and graphical interface. Sect. 4 gives a simplified verification exercise, and Sect. 5 demonstrates the framework on a reference structure. Sect. 6 closes by identifying next steps.

## 2 Methods

### 2.1 Spectral data decomposition

Spectral data can come from various sources, including experiments, computational fluid dynamics, and lower-fidelity methods. Regardless of the source, the objective is to reduce the time series into a non-dimensional spectral form that can be scaled and used for other flow conditions. The primary non-dimensional number of interest is the Strouhal number, defined in Eq. (1), with the frequency in hertz as $f$, the inflow velocity in meters per second as $V_{\text{inf}}$, and the characteristic length in meters as $L$. Once a frequency is non-dimensionalized by Strouhal number, it can then be scaled for different characteristic lengths and inflow velocities.

$$St = \frac{fL}{V_{\text{inf}}} \tag{1}$$

For non-circular shapes, the characteristic length is taken as the flow-normal projection of the major axis. Though we do not discuss it in detail, this characteristic length was chosen over the total planform projection (including thickness of the secondary axis), as it produced less variation in Strouhal number across wide AoAs.

Additionally, we define the lift and drag coefficients, which are necessary to scale the force amplitudes independent of the Strouhal frequency scaling. These are defined in Eq. (2) and Eq. (3), with the lift $L$, drag $D$, fluid density $\rho$, planform area $S$ (typically chord times unit length), and freestream velocity $V_{\text{inf}}$.

$$C_L = \frac{L}{0.5\rho S V_{\text{inf}}^2} \tag{2}$$

$$C_D = \frac{D}{0.5\rho S V_{\text{inf}}^2} \tag{3}$$

Next, in order to move from the time domain to the frequency domain, we can conduct a standard discrete Fourier transform like that shown in Eq. (4). In this transform, $X_k$ is the complex frequency domain coefficient, with the real part as the amplitude (properly scaled in Eq. (5), accounting for negative frequencies), and the imaginary as the phase. Then, $x_n$ is the time-domain



sample data and $N$ the total number of samples. It should be noted that the mean of the signal is included as the $k=0$ position, and is reconstructed likewise in this single-sided real-value analysis.

$$X_k = \sum_{n=0}^{N-1} x_n e^{-i \frac{2\pi}{N} kn}, \quad k = 0, 1, 2, \ldots, N/2 \tag{4}$$

$$A_k = \frac{2|X_k|}{N} \tag{5}$$

Equation (6) is the frequency vector corresponding to the $X_k$ coefficients, with the frequency $f_k$ in hertz, and the sampling

frequency (inverse of the time step) $f_s$.

$$f_k = \frac{k}{N} f_s, \quad k = 0, 1, 2, \ldots, N/2 \tag{6}$$

Equation (7) is the inverse of Eq. (4) (written using the cosine notation as opposed to the complex exponent), which can be used to reconstruct a signal after the amplitudes and frequencies are scaled.

$$x_n = \frac{1}{N} \sum_{k=0}^{N-1} |X_k| \cos\left(\frac{2\pi kn}{N} + \phi_k\right), \quad k = 0, 1, 2, \ldots, N/2 \tag{7}$$

However, since we are interested in identifying the major shedding frequencies, a reliable way to sort the transform is needed. Sorting by amplitude alone provides mixed results, while sorting by power spectral density (PSD) clearly differentiates between areas of high energy content.

$$PSD_k = \begin{cases} \dfrac{1}{N f_s} |X_k|^2, & k = 0 \text{ or } k = N/2, \\[2mm] \dfrac{2}{N f_s} |X_k|^2, & k = 1, 2, \ldots, N/2 - 1 \end{cases} \tag{8}$$

Additionally, due to the size of the problem across a wide range of AoAs, Reynolds numbers, and different shapes, we need

a reliable way to reduce the data without sacrificing accuracy. Keeping only the first several hundred sorted frequencies enables a reconstruction of accurate signals with reduced memory.

Putting this frequency space analysis into practice, Fig. 1 shows a sample lift and drag time signal and PSD frequency plot. The frequency content for the sample NACA 0018 data at 164° (16° reversed flow) AoA shows multiple peaks around 0.25 Hz and 1.25 Hz, corresponding to Strouhal numbers of around 0.02 and 0.09. These numbers are markedly different from the

flat-plate experimental values referenced and highlight the importance of this work for structures that could experience these



conditions.

**(a)** **(b)**

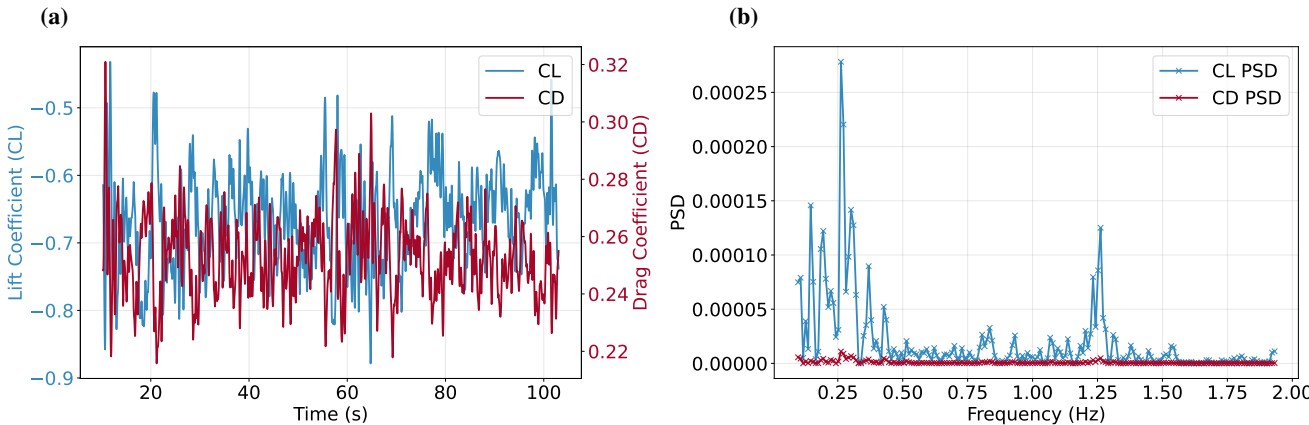

**Figure 1.** Sample time domain (a) and frequency domain (b) lift and drag data for a 1 m chord NACA 0018 airfoil at 164° AoA, an inflow of 3.75 ms$^{-1}$, and a Reynolds number of 5e+05.

The PSD shown in Fig. 1 indicates that the drag has similar frequency content as the lift but at much lower amplitudes. As the AoA approaches 90°, drag dominates, and for some intermediate conditions, different frequency harmonics show up in the lift versus drag. This is illustrated in Fig. 2, which presents the same information as Fig. 1 but for an AoA of 84°. Therefore, the 2-norm of the lift and drag is useful for sorting the peak Strouhal numbers and also enables a reduction of the dataset for computational memory without sacrificing reconstruction accuracy. Additionally, the combined force enables a more efficient overlap identification strategy, as even minor alignment has the potential to excite a modal frequency.

**(a)** **(b)**

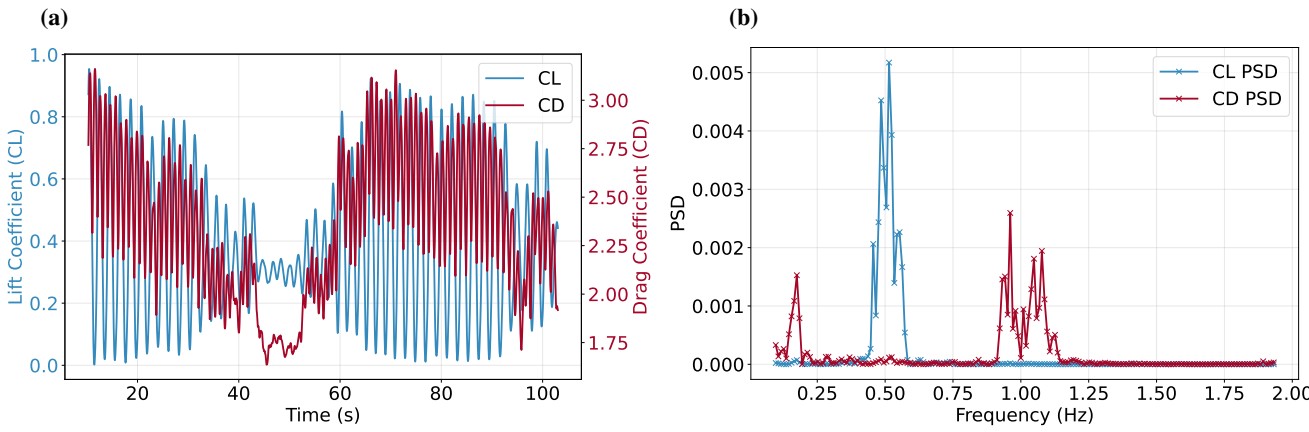

**Figure 2.** Sample time domain (a) and frequency domain (b) lift and drag data for a 1 m chord NACA 0018 airfoil at 84° AoA, an inflow of 3.75 ms$^{-1}$, and a Reynolds number of 5e+05.

By running this analysis for all AoAs (Fig. 3), we can extract all the frequency content, normalize by the Strouhal number, and retain the amplitude and phase information. The figure shows the first 30 frequencies, sorted by the 2-norm of the lift and





drag coefficients, termed CF for force coefficient. The multiple harmonics are clearly shown. All frequencies were tuned by increasing the Strouhal number by 0.07 for all Strouhal numbers to better align with other datasets. While accurate airfoil-specific datasets are in progress, the intent of this study is to focus on the methods and impacts for a representative dataset.

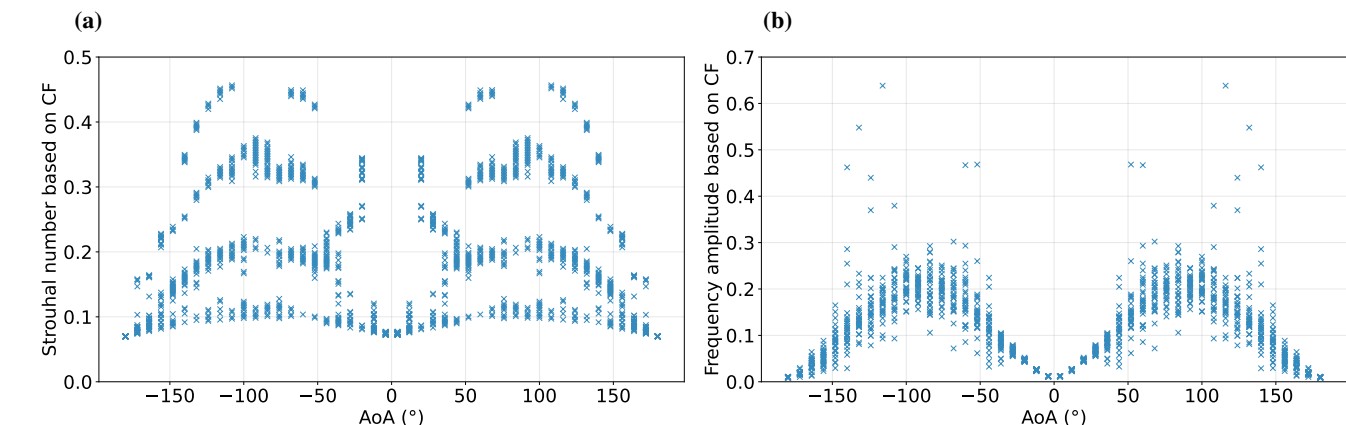

**Figure 3.** Strouhal number (a) and frequency amplitude (b) for the first 30 sorted frequencies based on the 2-norm of the sample lift and drag data for a 1 m chord NACA 0018 airfoil at an inflow of $3.75 \text{ ms}^{-1}$ and a Reynolds number of 5e+05. Note that all frequencies were tuned by increasing the Strouhal number by 0.07 for all Strouhal numbers.

With a complete dataset, we can now use the new geometry and flow conditions with the inverted Strouhal (Eq. (1)), lift coefficient (Eq. (2)), and drag coefficient (Eq. (3)), and with the inverse Fourier transform equation (Eq. (7)) to generate a scaled signal. This process enables independent scaling of both the shedding frequencies and the lift and drag for use in identifying potential shedding-modal frequency overlap, as well as for reconstructing time-domain force signals for one-way coupling.

**2.2 Linear algebra for multi-body AoA calculation**

To calculate the local AoA and local velocity vector, we use basic linear algebraic rotations and expressions, rotating the inflow vector about the prescribed rotation axis. The inflow velocity vector in the local coordinate frame of each blade element is obtained by rotating the freestream velocity $\mathbf{V}_\infty$ about the rotor axis. Specifically, the rotated inflow vector is given by

$$V_{\text{in,rot}} = R(\hat{a}_{\text{rot}}, -\psi_j) V_\infty \tag{9}$$

where $\hat{a}_{\text{rot}}$ is the unit vector defining the rotation axis of the rotor or component, and $\psi_j$ is the azimuthal position angle for index $j$. The operator $\mathbf{R}(\hat{a}_{\text{rot}}, -\psi_j)$ denotes a rotation matrix that transforms the global velocity vector into the local frame fixed to the blade element.

Figure 4 provides a pictorial view of the normal and tangential vectors on each multibody component, with an example inflow velocity vector in blue and rotation vector in black, with the objective to calculate the local static AoA inflow velocity
in the blade frame of reference for a given parked state.





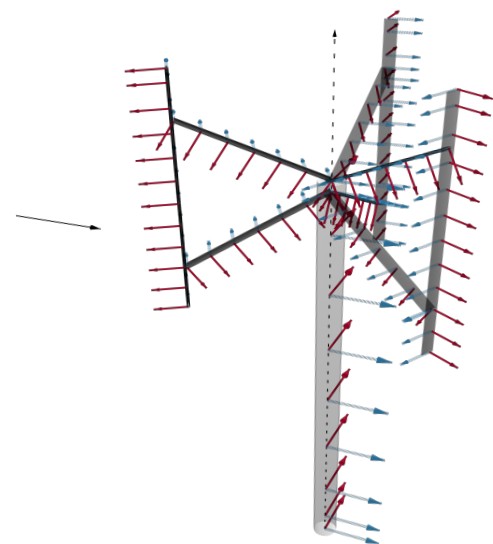

**Figure 4.** Example built up geometry in semi-transparent grey, with normal vectors in red, tangential vectors in blue, the rotation axis in dashed black, and the inflow vector in solid black. This information is used to calculate the local inflow and AoAs at every nodal position for all conditions.

At each evaluation point along the blade, the chord direction normalized vector $\hat{\mathbf{c}}$ and the normal direction vector $\hat{\mathbf{n}}$ are defined from the local geometry, which is inherent to a given multi-body design. Their corresponding unit vectors are representing the local chordwise and normal directions, respectively.

The inflow velocity $V$ components projected onto these directions are then computed as

$$V_c = V_{\text{in,rot}} \cdot \hat{c} \tag{10}$$

$$V_n = V_{\text{in,rot}} \cdot \hat{n} \tag{11}$$

corresponding to the chordwise and normal components of the local inflow velocity.

The local geometric AoA $\alpha$ is defined from these components as

$$\alpha = \tan^{-1}\left(\frac{V_n}{V_c}\right) \tag{12}$$

where the two-argument form of the arctangent is used in implementation to ensure correct quadrant identification.

Finally, the magnitude of the effective local inflow velocity $V_{\text{eff}}$ in the two-dimensional chord–normal plane is expressed as





$$V_{\mathrm{eff}} = \sqrt{V_n^2 + V_c^2} \tag{13}$$

This effective speed neglects any spanwise velocity component and represents the local magnitude of the relative flow that governs the aerodynamic lift and drag (including the shed frequency scaling) experienced by the section.

## 2.3 Minimizing distortion and complexity when interpolating

With data preprocessed and sorted by power spectral density, the interpolation space is two-dimensional across Reynolds number (from the local inflow) and AoA. Interpolation can be done for amplitude, phase offset, and Strouhal number, for each of the lift, drag, and moment coefficient datasets. Since interpolation is done for each sorted level, a limit on the number of sorted levels used can minimize the computational cost. Since the peak power spectral density tends to smoothly change with AoA, we assume that a clean dataset will inherently limit distortion from interpolation, and a fine enough grid can enable the much less costly linear interpolation methods to be used.

## 2.4 Combinatorial overlap reduction

Since the problem is combinatorial in nature, we address the combinations via nested for-loop logic as shown in Algorithm 1. This reduction must be done for all inflow conditions for every node in the structure, and compare against each modal frequency, harmonic, and level depth of sorted shedding frequency. Without care in the discretization choices, this $n^7$ problem (assuming the same discretization for each level, for simplicity), can explode quickly.

---

**Algorithm 1** Combinatorial Reduction

---

1: **for** each Inflow **do**
2:    **for** each Azimuth **do**
3:       **for** each Component **do**
4:          **for** each Node **do**
5:             Calculate Local Velocity & AoA
6:             Interpolate Strouhal Numbers, CL & CD Amplitudes, and Phases between Chord Reynolds and AoA
7:             Dimensionalize to Frequencies, Lifts, Drags
8:             **for** Each Modal Frequency **do**
9:                **for** each Modal Harmonic **do**
10:                   **for** each Shedding Frequency **do**
11:                      Save Case if Worst & Above Filter Thresholds
12:                      Reconstruct Forcing Signals from Frequency, Lift, Drag, and Phase data
13:                   **end for**
14:                **end for**
15:             **end for**
16:          **end for**
17:       **end for**
18:    **end for**
19: **end for**

---





## 2.5 Signal reconstruction and load synthesis

Given the discrete amplitude spectrum $A_k$ and phase angles $\phi_k$ at frequencies $f_k$, the time-domain coefficient $c(t)$ (e.g., $C_L$) can be reconstructed via the inverse Fourier synthesis (Oppenheim and Schafer, 2010):

$$c(t) = A_0 + \sum_{k=1}^{N/2} A_k \cos(2\pi f_k t + \phi_k) \tag{14}$$

where $A_0$ represents the mean component. This reconstruction is done for all of the forces (lift, drag, and moment). For real-valued signals, conjugate symmetry is enforced to ensure real reconstruction. Turbine-level thrust, torque, and moment time signals can be obtained by aggregating per-segment spectral contributions according to their local geometry and reference areas and integrating.

## 3 Software implementation (VorLap)

VorLap is an open-source Python framework designed to predict vortex-induced vibration risks in static or very-low-speed rotating structures using spectral flow data derived from computational fluid dynamics or experimental measurements. The software implements the algorithms discussed in the methods section and provides utilities for file IO, visualization, and a graphical user interface. While the example here is for a vertical-axis wind turbine, any application could be simulated, such as communication towers, underwater structures, horizontal-axis wind turbines, etc. The following subsections describe the core modules of the software and their roles in the overall workflow.

### 3.1 FileIO

The file IO module handles ingestion of structural geometry and spectral airfoil datasets from standardized file formats. Structural components are defined through CSV files containing a two-header format. The first header row specifies component-level metadata (identifier, translation, rotation, pitch) with the corresponding values in the following row. The second header row specifies segment-wise geometry data, including spatial coordinates $(x, y, z)$, aerodynamic parameters (chord, twist, thickness, offset), and airfoil assignment with the following rows each containing the values for an individual segment. Airfoil assignments point to an airfoil CSV file that contains the non-dimensional coordinates of the specified airfoil shape in two columns. Spectral airfoil data are loaded from HDF5 files containing three-dimensional arrays organized as [Reynolds number × AoA × Frequency] for Strouhal numbers, amplitudes, and phases of lift, drag, moment, and combined force coefficients.

The module also provides functionality for exporting computed force time series to CSV format, with structured headers indicating node positions and force components. This CSV intermediate file enables direct integration with downstream structural analysis tools for one-way coupling simulations.





## 3.2 Computation

The computational module implements the linear algebra transformations described in Eq. (9)–Eq. (13) to compute local AoAs and effective velocities for each structural segment across all azimuthal orientations and inflow conditions. The core computation follows the combinatorial reduction algorithm outlined in Algorithm 1, iterating through inflow speeds, azimuthal positions, components, and nodes to evaluate the complete parameter space.

The module also handles force vector transformations, rotating local lift and drag forces into the global coordinate frame using Rodrigues' rotation formula. This formulation enables computation of aggregate forces and moments at the structure level, which are accumulated across all components and segments.

Additionally, the module implements signal reconstruction functionality using the inverse Fourier transform approach described in Eq. (14). This synthesizes time-domain force signals from the interpolated spectral data, enabling generation of force time series at specified structural nodes for selected operating conditions. The reconstruction handles mean values separately from oscillatory terms and includes Nyquist frequency validation to prevent aliasing artifacts.

## 3.3 Visualization

The visualization module generates three-dimensional visualizations of structural geometry, including component placement, chord vectors, normal vectors, rotation axes, and inflow directions. The visualization uses Plotly for interactive 3D rendering, enabling users to inspect geometry from multiple viewpoints and validate component configurations before analysis execution.

The visualization serves both as a validation tool (ensuring geometry is correctly specified) and as a communication aid (illustrating structure configurations in presentations and documentation). The module includes fallback rendering options for headless computing environments, ensuring compatibility with automated workflows and continuous integration systems.

## 3.4 Graphical user interface

To improve accessibility for users without programming experience, VorLap includes a graphical user interface (GUI) implemented in Python's Tkinter framework. The interface provides an integrated workflow from data loading through results visualization, organized into two tabs:

**Simulation Setup Tab:** Handles parameter configuration through form-based inputs for fluid properties, inflow conditions, frequency analysis settings, and structural component loading. The interface includes real-time validation and error messaging to guide users through proper configuration. The GUI can receive input manually by editing the fields in the application or by uploading files. Uploaded files can also be modified before executed.

**Plots & Outputs Tab:** Provides interactive visualization of analysis results with embedded matplotlib plotting capabilities. Users can switch between different visualization modes (frequency overlap, force components, moment components) and export plots directly from the interface. The tab includes controls for adjusting plot parameters and saving results in standard image formats.





The GUI maintains full functional parity with the command-line interface, ensuring that all analyses can be reproduced through either pathway. Results are automatically saved to user-specified directories with organized folder structures for plots, time-series data, and configuration files.

# 4   Verification

To verify the tool is implemented without errors, we exercise the full VorLap workflow for a single blade section. The verifica-
tion focused on two aspects: (1) spectral reconstruction accuracy, ensuring that frequency and amplitude content are preserved throughout the workflow, and (2) correct identification of shedding–modal frequency overlap within the discretized parameter space.

## 4.1   Reconstruction accuracy

The force time series from a single blade section was decomposed and reconstructed using VorLap. Data from Bidadi et al.
(2024) were preprocessed as described in Sect. 2, then passed through the VorLap tool, including transformation into the frequency domain, filtering to remove uncertain low-amplitude harmonics, and reconstruction into the time domain. Figure 5 shows the reconstructed dimensional force signal compared to the reference data for the FFA-W3-211 blade with a freestream velocity of 75 ms$^{-1}$, a chord length of 1 m, an AoA of -144° (216°), corresponding to reversed-flow high AoA conditions. VorLap reconstruction reproduces both the dominant frequency content and the amplitude envelope with negligible phase shift,
confirming that the inverse Fourier processing and scaling methods are quantitatively accurate compared to the original data. Minor differences at lower frequencies correspond to harmonics below the user-defined amplitude threshold, which are intentionally filtered for computational efficiency. The reconstructed signal is an acceptable representation of the original data.

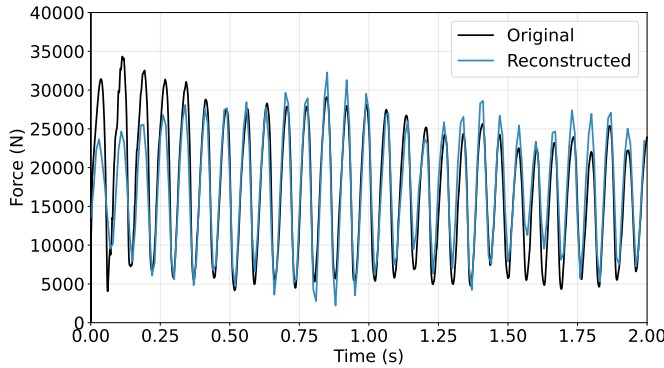

**Figure 5.** Original and reconstructed force time series for a single blade after VorLap processing and scaling. The reconstructed signal shows good amplitude (within 5 %) and phase (within 1 %) alignment.



## 4.2 Frequency mapping verification

To confirm the frequency scaling methods, the scaled spectrum was compared against a modal frequency of 15 Hz, chosen intentionally to match one of the dominant frequencies present in the airfoil force data. This verification was done for a variety of inflow angles, speeds, and high-pass filtering. The heatmap shown in Figure 6 gives the results from VorLap, verifying that the dimensionalization via Strouhal scaling correctly adjusts shedding frequencies with changing inflow velocity. The peak overlap identified by VorLap occurred at the expected angle and velocity combinations, consistent with the known dominant

shedding modes near 15 Hz for the 216° AoA case. This verification demonstrates correct implementation of frequency scaling and interpolation and gives us greater confidence as we move to the full structure demonstration.

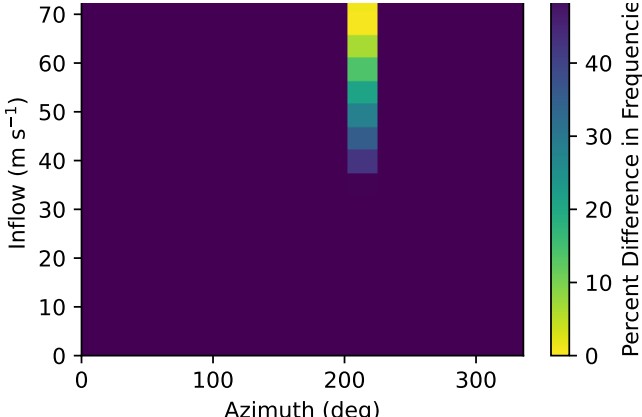

**Figure 6.** Heatmap showing the worst case overlap of a 15 Hz modal frequency and scaled shed frequencies, including high-pass filtering. The expected overlap at 216° AoA is highlighted.

## 5 Demonstration with example structure

The motivation in Sect. 1.1 showed a simplified approach to identify that there was a potential for vortex-induced vibration

onset during parked conditions for the reference H-VAWT in Moore et al. (2024) (geometry shown in Fig. 4). In this section, we exercise the VorLap tool to refine the identification of potential areas of overlap and then demonstrate sample forcing signal reconstruction.

It is important to reiterate that different environmental conditions, airfoils, and fluid scaling (Reynolds number), may necessitate using vortex shedding datasets that are for the application. Therefore, for this paper, we present an architecture and method

for how to navigate this highly dimensional problem and properly scale both the frequency and forces, but we recommend using datasets that appropriately match the application space.





## 5.1  Model setup

For this example demonstration, the turbine analysis was done with the configuration shown in Table 1 and using the example computational fluid dynamics (CFD) dataset previously discussed. The demonstration is intended to simulate a wide variety of possible vortex shedding frequency overlap where shedding may occur (a distributed wind turbine during normal parked conditions). We do not simulate high wind speed because it is anticipated that the turbulence intensity will dominate the loading and likely disrupt vortex shedding (Blackburn and Melbourne, 1996).

**Table 1.** Reference turbine and simulation definition.

| Name | Value |
| --- | --- |
| Diameter | 10.5 m |
| Blade & Strut Chord | 0.5 m |
| Blade Height | 10.54 m |
| Blade Airfoil | NACA 0018 |
| Struts per Blade | 2 |
| Strut Angle | 30° |
| Wind Vector | (1,0,0) |
| Rotation Vector | (0,0,1) |
| Rotation Axis Offset | (0,0,0) |
| Fluid Density | 1.225 kgm$^{-3}$ |
| Fluid Dynamic Viscosity | 1.81E-5 Pa-s |
| Azimuth Range | 0:5:120° |
| Inflow Velocity Range | 1:1:16 ms$^{-1}$ |
| Number of Modal Harmonics | 1 |
| Sorted Frequency Comparison Depth | 10 |
| Amplitude Coefficient Minimum Cutoff | 0.5 |

The azimuth range was sampled in 5° increments and restricted to 120° due to symmetry (noted below). The turbine rotates about a vertical axis, so either wind speed or parked azimuth could change the relative azimuth angle. Since the tool was designed to be application agnostic, the frame of reference for azimuth follows standard mathematical notations (0° is at the east position). To align with vertical-axis wind turbine notation (0° at the north position), the input geometry was simply defined as rotated by 90°.

Using the structural definition and solver in (Moore et al., 2024), the first natural or modal frequencies for the structure are 2.711, 6.021, 7.151, 7.689, 7.863, and 11.615 Hz. The shapes are visually depicted in Fig. 7. Of note is that the third mode is a series of combinatorial variations of the three blades bending in and out of sync with one another, spanning 7.151, 7.689, and 7.863 Hz. These natural frequencies, and their respective harmonics, are what we compare the shedding frequencies to.




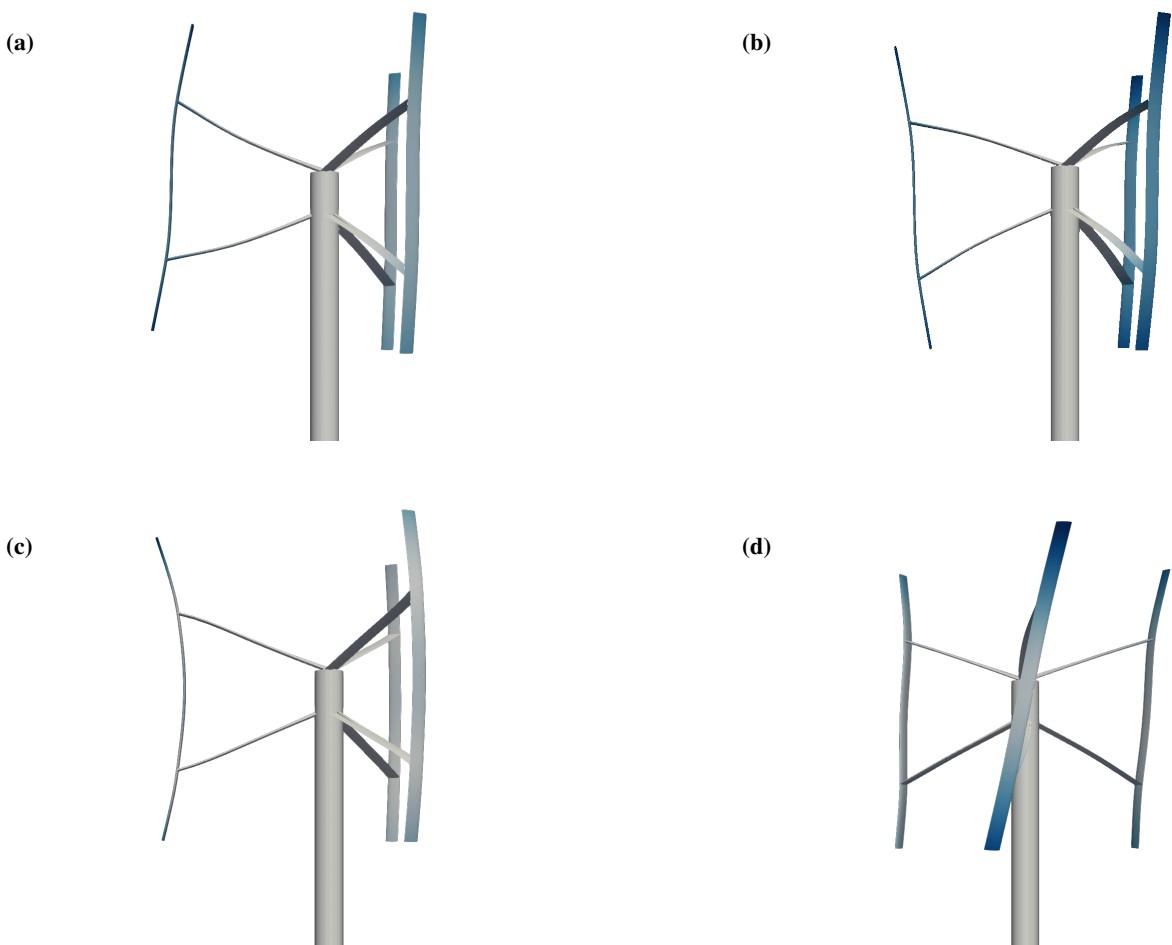

**Figure 7.** Mode shapes and frequencies for the reference turbine, using the structural definition and solver in Moore et al. (2024). First mode (a), associated with 2.711 Hz. Second mode (b), associated with 6.021 Hz. Third mode (c), associated with the family of 7.151, 7.689, and 7.863 Hz (different combinations of blade bending in or out of sync). Fourth mode (d), associated with 11.615 Hz.

## 5.2 Frequency overlap examples and discussion

While the example structure may experience any direction of inflow velocity during parked conditions due to flow direction change, the turbine may tend to stop at zero-torque azimuthal positions across the full 360° azimuth range more frequently. However, due to the three blade and arm sets being fully symmetrical, we can focus on a range up to 120° azimuth.



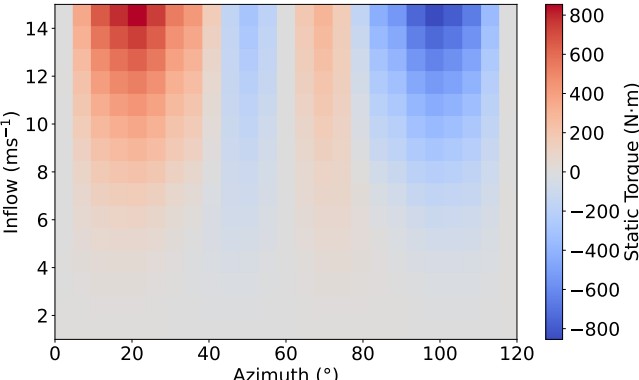

**Figure 8.** Static torque for the reference turbine about the axis of rotation showing low torque bands around 0°, 60°, and 120°, as well as 45° and 75° to a lesser extent.

Figure 8 shows the static torque for the reference turbine about the axis of rotation for azimuths of 0° to 120°. Low torque positions include 0°, 60°, and 120°, and to a lesser extent 45° and 75°. All positions need to be considered, but these positions provide possible higher probability parked configurations for this particular application.

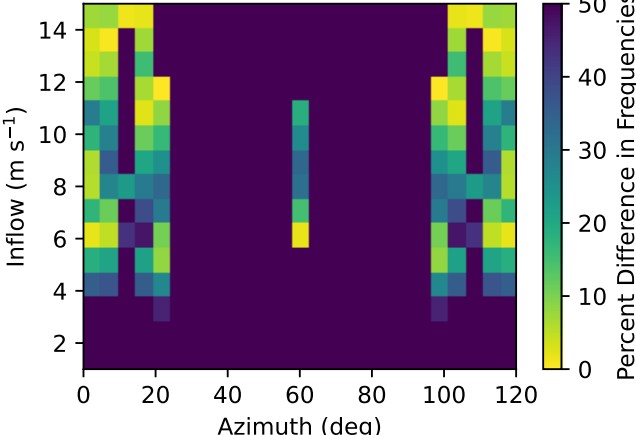

**Figure 9.** Heatmap showing the percent difference between the reduced combinatorial set, including filtering thresholds, of modal frequencies and scaled shed frequencies.

With these higher-probability positions in mind, we now consider the heatmap in Fig. 9 showing the case with the highest frequency overlap. The worst-case amplitude and frequency overlap scenario is shown because the coefficient filter is set high enough that only the highest-amplitude cases are being considered in the combinatorial reduction. For designers using this tool, it is recommended that they consider multiple different filtering levels to understand problem areas for their specific designs

and establish logical thresholds based on structural damping, etc.

The first observation from this demonstration is that the high-level analysis done at the beginning of the paper appears to have been correct, but that a high degree of overlap risk (the bright yellow areas) is isolated to specific conditions. The second





observation is that the results appear to be highly symmetric about 60° azimuth. This symmetry would appear to be in large part to the thin, non-cambered profile of the NACA 0018 acting similar to a flat plate during high AoAs. Referring back to Fig. 3, the variations in shedding frequency appear to become non-symmetric between forward and reversed flow only at lower relative AoAs. However, for those conditions, the shedding amplitude drops off sharply and is filtered out. For a thicker or cambered airfoil, these trends would likely be different, but for this study with a high degree of symmetry due to the airfoil geometry, we may further refine the focus to between 0° and 60°.

Focusing on azimuth angles between 0° and 60°, Table 2 shows a few example summaries of high-risk overlap conditions and where they occur. With the filtering so selective, the cases highlighted are limited to the peak amplitude of the input data, and a designer would need to determine if this threshold is appropriate for the structure. The trends shown tend to favor higher AoAs until flow reversal effectively creates a shallow AoA again. Also, for this case, the higher-frequency modes tend to be overlapped only with higher velocity flow, so they may not be seen often for this type of structure except during certain maintenance conditions. For example, if a turbine was parked for maintenance and the wind speed increased above the normal operation cut-in, then a higher-frequency mode might become excited.

**Table 2.** Sample high-risk overlap summaries for the first modal harmonic. Note the fourth mode was dominated by overlap with the other modes for this case.

| Mode | Harmonic | Azimuth | Inflow | % Difference | Location | AoA | Chord Reynolds |
|---|---|---|---|---|---|---|---|
| 1 | 1 | 0° | 6 ms$^{-1}$ | 1.9 % | Blade 3 | 120° | 203k |
| 1 | 1 | 60° | 6 ms$^{-1}$ | 1.6 % | Blade 3 | 60° | 203k |
| 1 | 1 | 0° | 8 ms$^{-1}$ | 5.6 % | Arm 2 | 140° | 179k |
| 2 | 1 | 20° | 12 ms$^{-1}$ | 0.13 % | Blade 2 | -140° | 406k |
| 2 | 1 | 5° | 15 ms$^{-1}$ | 7.7 % | Blade 3 | 115° | 507k |
| 2 | 1 | 60° | 11 ms$^{-1}$ | 18.4 % | Blade 3 | 60° | 372k |
| 3 | 1 | 0° | 15 ms$^{-1}$ | 7.1 % | Blade 3 | 120° | 507k |
| 3 | 1 | 10° | 15 ms$^{-1}$ | 1.3 % | Blade 2 | -130° | 507k |
| 3 | 1 | 0° | 15 ms$^{-1}$ | 1.8 % | Blade 3 | -135° | 507k |

Also of note is that while the blades tend to dominate the samples chosen, other components can begin to overlap. In the Arm 2 overlap sample case, the local velocity that the shedding frequency is scaled with is in the local blade frame of reference (note Sect. 2.2), which, with a skew angle, decreases the inflow and in turn the shed frequency. This factor changes the range of predicted frequencies being shed, leading to possible overlap. For high skew angles, this assumption may not hold, as shed vortices may tend to combine, changing the shed frequency (Zhang et al., 2023).

## 5.3 Signal reconstruction and discussion

For this demonstration, we choose the second-line case in Table 2. This condition has shedding alignment coming from the third blade, and since the blade is uniform and straight (as opposed to a different design with variable chord, twist, shape, etc.), the reconstructed signal for all of the nodes along the blade are the same, so we choose the second node. Now, as discussed in





Sect. 2.5, we use all the frequency and phase content for both the lift and drag at this interpolated AoA and Reynolds number. We recall that the data were scaled to the size and flow conditions for frequency via the Strouhal scaling, and the forces were scaled through lift and drag coefficient scaling.

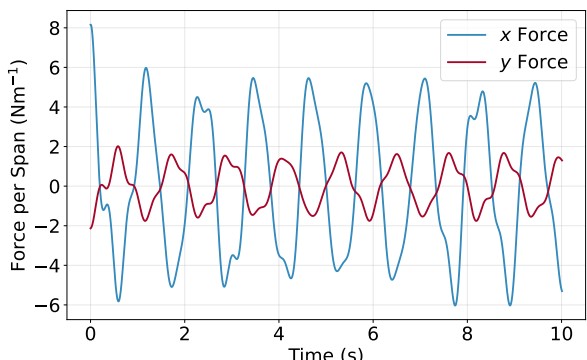

**Figure 10.** Sample reconstructed force time signals for Blade 3, Node 2 at the $60°$, $6\,\mathrm{ms}^{-1}$ turbine parked condition.

Figure 10 shows the sample reconstructed force time signals for Blade 3, Node 2 at the $60°$, $6\,\mathrm{ms}^{-1}$ turbine parked condition in the turbine global $x$ and $y$ directions. Using this same method for all defined nodes across all components with their respective conditions, a full set of structural loads can be created and can be used in one-way coupling with structural modeling software. These capabilities are automated in the VorLap tool, enabling designers to conduct further structural analyses. Additionally, a designer could iteratively export loads for differing inflow velocities and angles and combine them to create a loading profile

representative of long-term parked conditions.

## 6    Conclusions

This work presents a reproducible, open-source framework for evaluating vortex-induced vibration onset using spectral flow data, enabling systematic identification of overlap between vortex shedding frequencies and structural natural frequencies. The VorLap method and accompanying software establish a consistent means of scaling and interpolating high-dimensional

aerodynamic data spanning Reynolds number, AoA, and geometry to arbitrary multi-body structures. By combining spectral decomposition, linear algebraic transformations, and combinatorial reduction, the workflow bridges the gap between fundamental unsteady flow data and engineering design application. For a broad variety of structures, such as radio towers, underwater structures, horizontal-axis wind turbines, vertical-axis wind turbines, and others, designers can utilize VorLap to quickly identify, avoid, and include vortex-induced vibration considerations in their design processes.

Verification against published CFD datasets and reconstructed time-domain signals demonstrated that the approach preserves both frequency and amplitude fidelity through the complete data pipeline. Further verification and validation would be ideal in future work. Application to a reference vertical-axis wind turbine illustrated how specific vortex shedding frequencies can



align with modal responses under specific parked conditions, corroborating the analytical intuition outlined in the motivation section.

The results suggest that spectral lift and drag datasets, when treated through non-dimensional scaling and organized interpolation, can support early-stage design screening for the risk of vortex-induced vibration onset. Also, the open architecture of VorLap facilitates integration with existing structural solvers for one-way coupled load reconstruction or reduced-order modeling. Future work is anticipated to result in a high-quality CFD dataset and improvements to skew corrections.

*Code availability.* The VorLap code can be found at https://github.com/sandialabs/VorLap, with the verification case and demonstration
included as part of the code test set.

*Author contributions.* **Conceptualization:** KRM; **Methodology:** KRM, KLB; **Software:** KRM, KLB; **Validation:** HKR, KRM; **Visualization:** KRM, KLB, HKR; **Writing, original draft:** KRM; **Writing, review & editing:** KRM, HKR, KLB; **Supervision/Project administration/Funding acquisition:** BJS

*Competing interests.* The authors declare that they have no conflict of interest.

*Disclaimer.* Sandia National Laboratories is a multi-mission laboratory managed and operated by National Technology & Engineering Solutions of Sandia, LLC (NTESS), a wholly owned subsidiary of Honeywell International Inc., for the U.S. Department of Energy's National Nuclear Security Administration (DOE/NNSA) under contract DE-NA0003525. This written work is authored by an employee of NTESS. The employee, not NTESS, owns the right, title and interest in and to the written work and is responsible for its contents. Any subjective views or opinions that might be expressed in the written work do not necessarily represent the views of the U.S. Government. The
publisher acknowledges that the U.S. Government retains a non-exclusive, paid-up, irrevocable, world-wide license to publish or reproduce the published form of this written work or allow others to do so, for U.S. Government purposes. The DOE will provide public access to results of federally sponsored research in accordance with the DOE Public Access Plan.

This work was authored in part by the National Laboratory of the Rockies for the U.S. Department of Energy (DOE), operated under Contract No. DE-AC36-08GO28308. Funding provided by U.S. Department of Energy Office of Critical Minerals and Energy Innovation Wind
Energy Technologies Office. The views expressed in the article do not necessarily represent the views of the DOE or the U.S. Government. The U.S. Government retains and the publisher, by accepting the article for publication, acknowledges that the U.S. Government retains a nonexclusive, paid-up, irrevocable, worldwide license to publish or reproduce the published form of this work, or allow others to do so, for U.S. Government purposes.





*Acknowledgements.* This work has been funded by the United States Department of Energy (DOE) Wind Energy Technologies Office under

the Distributed Wind Aeroelastic Modeling (dWAM) project. AI tools were used in the conceptualization and rough drafting phases of the

method, software, and paper. However, due to the high AI error rate, all AI generated content was manually rewritten or extensively reviewed

and modified to produce draft and final products.



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
