# Peer review of "Method for using spectral flow data to predict vortex-induced vibration onset of static structures"

_Wind Energy Science, 2025_

## Referee Comment (RC1)

This paper presents a structured framework for identifying vortex-shedding–induced vibration risk using spectral aerodynamic data, with a particular focus on parked or quasi-static wind turbine configurations. The methodology is logically developed from aerofoil spectral inputs through geometric transformation, Strouhal-based frequency scaling, and combinatorial reduction and used in turbine-level frequency-overlap assessment and force reconstruction. The introduction of the open-source VorLap tool is a good contribution, and the verification and demonstration cases provide convincing evidence. Overall, the approach is sound and useful as a screening and load-generation tool for early-stage design and risk assessment. Please refine the manuscript or provide additional clarification addressing the following comments to improve transparency, interpretation, and reproducibility of the proposed methodology:

1- In Eq. 1 and Eq. 2 L is used for two different parameters (Lift and characteristic length). Please revise accordingly.

2- Using suffixes such as $V_{\text{inf}}$ for a parameter is not recommended. Please use Greek/Roman symbols e.g., $V_\infty$.

3- Please use capital H in 'hertz' or 'Hz' across the paper.

4- Please use half-space before all your Hz, to make sure your unit stand with the number on the same line.

5- You must clearly state where the data used in Fig. 1-Fig.3 are coming from? A proper citation needs to be used.

6- Can you explain that in Fig. 1, have you removed the mean values before calculating the PSDs? If not, how did you manage the large PSD values near zero frequencies?

7- In Page 6, you mentioned that you have moved the St values by 0.07 to adjust to other data sets. Firstly, what datasets? Please clarify or maybe present those data sets in the corresponding graph. Secondly, what would be the impact of this adjustment in VIV prediction?

8- In Page 6, please clarify that you have used $\sqrt{C_L^2 + C_D^2}$ to calculate PSD amplitudes and frequencies.

9- In Line 123 you have used $V_\infty$ while earlier you used $V_{\text{inf}}$. Are they two different parameters?

10- In Fig. 3 you mention that Re=500000 but with c=1 and $V_\infty = 3.75$ , Re=250000. Please clarify what is gone wrong?

11- In Sec. 2.2, please clarify whether the local inflow velocities and angles of attack are evaluated for a static (parked) turbine or under rotation?

12- I suggest you add parameters (such as incoming flow velocity, Vn, Vc etc) to the vectors shown in Fig4.

13- In many parts of the paper, a language is used that looks difficult to understand for the reader, even though the method is correct, it is not explained plainly. I would strongly suggest that the authors use a simple language. For example, in line 145-150, what is 'sorted level'?

14- Algorithm 1 describes a nested brute-force search with filtering. Please clarify the criteria used to define 'worst' cases and the 'thresholds' applied for overlap reduction or clarify these in the text.

15- The paper would benefit from an explicit statement early in Section 2 explaining why force reconstruction is required and how it enables VIV screening. Currently, the logic for synthesizing time-domain forces from CL/CD/St data must be speculated by the reader. Additionally, it would be helpful to clarify what new information is gained from the reconstructed force signals relative to direct spectral frequency matching?

16- The heatmaps in Figures 9 and 6 show minimum frequency mismatch persisting at higher inflow velocities. Please clarify why this behaviour occurs?  Does this arise from the

consideration of multiple shedding harmonics? This is a bit misleading since VIV usually occurs at a certain velocity and then disappears at higher velocities.

17- The present analysis identifies frequency proximity but does not account for structural damping or resonance bandwidth. It would be helpful to explicitly state this limitation and clarify how designers should interpret percent frequency difference in the context of damping and lock-in behaviour.

18- The reconstructed forces represent rigid-body vortex shedding excitation and do not include motion-induced VIV forces or lock-in effects (e.g., aerodynamic damping). Please clarify these in the text.